# Preventive Therapies in Peripheral Arterial Disease

**DOI:** 10.3390/biomedicines11123157

**Published:** 2023-11-27

**Authors:** Aangi J. Shah, Nicholas Pavlatos, Dinesh K. Kalra

**Affiliations:** 1Department of Internal Medicine, University of Louisville School of Medicine, Louisville, KY 40202, USA; aangi.shah@louisville.edu (A.J.S.); nicholas.pavlatos@louisville.edu (N.P.); 2Division of Cardiology, University of Louisville School of Medicine, Louisville, KY 40202, USA

**Keywords:** peripheral artery disease, lipids, atherosclerosis, prevention, major adverse cardiovascular events, lipid-lowering therapy, antithrombotic therapy, hyperglycemia management, hypertension management, growth-factor therapy, statins, PCSK9 inhibitors, ticagrelor, rivaroxaban, clopidogrel

## Abstract

Atherosclerosis, while initially deemed a bland proliferative process, is now recognized as a multifactorial-lipoprotein-mediated inflammation-driven pathway. With the rising incidence of atherosclerotic disease of the lower extremity arteries, the healthcare burden and clinical morbidity and mortality due to peripheral artery disease (PAD) are currently escalating. With a healthcare cost burden of over 21 billion USD and 200 million patients afflicted worldwide, accurate knowledge regarding the pathophysiology, presentation, and diagnosis of the disease is crucial. The role of lipoproteins and their remnants in atherosclerotic vessel occlusion and plaque formation and progression has been long established. This review paper discusses the epidemiology, pathophysiology, and presentation of PAD. PAD has been repeatedly noted to portend to poor cardiovascular and limb outcomes. We discuss major therapeutic avenues for the prevention of major cardiovascular adverse events and major limb adverse events in patients with PAD.

## 1. Introduction

The American Heart Association’s (AHA) 2021 guidelines call for a more precise definition of peripheral artery (or arterial) disease (PAD) and its definitional distinction from peripheral vascular disease, which may also include venous and lymphatic disease. These parameters define PAD as ‘lower extremity’ PAD, differentiating it from diseases involving other vascular territories, such as the carotid or renal vessels, and typically describe it as “the atherosclerotic obstruction from the aortoiliac to the pedal arteries” [1]. PAD continues to be a frequently missed diagnosis due to the lack of the characteristic claudication symptoms in all patients and other protean manifestations of the disease entity. In fact, only about 5–35% patients have the classical claudication symptoms. Up to 60% of patients with PAD are asymptomatic, and 10–50% of patients report atypical symptoms, such as leg pain that improves with continued walking or failing to resolve pain with rest [2]. A recent international survey involving 9098 respondents reported a low self-reported PAD awareness amongst the general public, with the awareness most restricted in the age groups over 45 years old [3].

The PARTNERS (PAD Awareness, Risk, and Treatment: New Resources for Survival) study conducted in 1999 included 150 primary care practices across the United States. The study reported that only 49% of physicians were aware of PAD diagnosis in the patients [4]. The most devastating complication is critical limb ischemia (CLI), which has been observed in about 11% of patients with PAD, entailing the risk of amputation if urgent revascularization is not performed [5]. Additionally, there are rigorous data suggesting increased all-cause mortality, cardiovascular mortality, and major coronary events in those patients with PAD as opposed to those without it [2]. A meta-analysis of sixteen epidemiological studies reported a reverse J-shaped curve relationship between the risk of death and rising ankle brachial index (ABI). The 10-year risk of cardiovascular mortality was almost 3–4 times higher with an ABI < 0.09 (18.7% in males and 12.6% in women). The study also reported the at least twice as many overall mortality, cardiovascular mortality, and major coronary event rates for all Framingham risk groups for ABI levels < 0.9 compared to 0.9–1.14 [6].

### 1.1. Epidemiology

It is estimated that the overall prevalence of PAD among those aged >40 years old in the United States is 4–7%, with an annual cost burden exceeding USD 21 billion [7,8]. Studies have shown that being diagnosed with PAD dramatically increases mortality with a 10-year all-cause mortality of 27% in the reference range, compared to 56% for asymptomatic PAD, 63% for those with intermittent claudication, and 75% for severe limb ischemia [9]. As is true with any atherosclerotic disease, age is one of the strongest risk factors for PAD. The prevalence is ~1% of those aged 40–49 and increases exponentially, doubling with each subsequent decade of life [10]. Men and women have similar prevalence of PAD; however, women commonly lack clinical manifestations of PAD and, as a result, are underdiagnosed and undertreated. As the population ages, females are more likely to have more severe manifestations of the disease, such as critical limb ischemia [11]. United States incidence rates are highest among African Americans, followed by Hispanic Americans, who are at a slightly higher risk than non-Hispanic Whites Americans [12]. Only half of patients with PAD exhibit lower extremity symptoms, as they suffer from other comorbidities that limit their mobility, such as cardiopulmonary disease or arthritis [13]. Other key risk factors for PAD include tobacco use, diabetes, hypertension, renal disease, hyperlipidemia, obesity, autoimmune diseases, HIV (human immunodeficiency virus), genetics, and low socioeconomic status [1].

### 1.2. Pathophysiology/Pathogenesis

Multiple factors contribute to the pathogenesis of atherosclerosis, including endothelial dysfunction, hyperlipidemia, inflammation, and plaque formation. The resultant hypoperfusion of distal tissues results in the genesis and proliferation of collateral vessels via angiogenesis and arteriogenesis.

Endothelial dysfunction is an initial step in the process of atherogenesis and is caused by chronic stressors, such as hyperlipidemia, hypertension, smoking, diabetes, and inflammation. Endothelial dysfunction is mediated by a reduction in the bioavailability of vasodilators, most notably nitric oxide (NO) [14]. The suboptimal bioavailability of NO is thought to be caused by high levels of low-density lipoproteins or LDL. Elevated LDL levels lead to the formation, progression, and subsequent complications of atherosclerosis. NO decreases endothelial permeability, thereby decreasing the flux of lipoproteins into the vessel wall, directly preventing the oxidation of said lipoproteins [15]. Additionally, NO mediates endothelium-dependent vasodilation by opposing the effects of vasoconstrictors such as endothelin and angiotensin II [16,17]. It also inhibits the proliferation of vascular smooth muscle cells, monocyte adhesion/ infiltration, and platelet–vessel wall interaction, all of which result in atherosclerosis [18].

Lipid abnormalities, specifically those with elevated concentrations of apoB-containing lipoproteins, play key roles in the formation of atherosclerosis. An early step of atherogenesis is the subendothelial retention of apoB-100 containing lipoproteins such as LDL. LDL retention in the vessel wall allows the oxidation of unesterified cholesterol and surface phospholipids [19]. The oxidation of LDL results in the formation of free radical products known as isoprostanes, which are isomers of conventional prostaglandins and key markers of oxidative stress in atherosclerosis. This process occurs in any cell within the artery, including macrophages, smooth muscle cells, endothelial cells, and T-lymphocytes [20]. Oxidized LDL promotes atherogenesis through multiple mechanisms such as acting as a chemoattractant for macrophages that ingest oxidized lipoproteins, transforming them into foam cells [21]. Cholesterol accumulation in foam cells leads to reduced macrophage mobility, thereby trapping macrophages in the vessel wall, leading to mitochondrial dysfunction, apoptosis, and necrosis. This process results in the release of prothrombotic molecules, cellular proteases, inflammatory cytokines, and oxidized LDL free radicals, all of which serve as irritants affecting the vessel wall and cause subsequent plaque formation [22,23].

Atherosclerotic plaque can progress over decades with fibrosis and calcification. Histologically, atherosclerosis initially presents as a fatty streak, which is a focal thickening of the intima via the accumulation of foam cells, intra/extracellular lipid deposits, extracellular matrix, and smooth muscle cells. As these lesions mature and expand, more smooth muscle accumulates in the intima, and it can undergo apoptosis, resulting in additional macrophage accumulation, aiding the transformation of a fatty streak to a plaque [24]. Intraplaque hemorrhage is a result of neovascularization and essential to plaque progression, instability, and, ultimately, rupture [25].

As a result of the hemodynamic changes occurring within the occluded artery, both angiogenesis and arteriogenesis play roles in providing distal tissue with adequate circulation. Arteriogenesis is initiated when there is increased blood flow through collateral arteries. The increased flow upregulates the endothelic nitric oxide synthase in endothelial cells, as well as the release of NO, promoting the dilation of collateral vessels [26]. Increased flow also results in the upregulation of adhesion molecules, which promote the recruitment of monocytes that differentiate into macrophages and release various extracellular matrix remodeling enzymes that help to increase the diameter of the collateral artery [27]. Angiogenesis refers to formation of new blood vessels from pre-existing capillary structures. This process is largely mediated by Hypoxia-Inducible Factor (HIF)-1, which is released as a response to poor oxygen delivery. HIF-1 stimulates vessel growth via multiple mechanisms, such as endothelial progenitor cells and smooth muscle cell recruitment, vascular permeability stabilization, and vessel maturation [28]. Figure 1 depicts the pathogenetic process of atherosclerosis.

The following list describes the contents of each of the images shown in Figure 1.

Normal vessel wall with endothelial structure and function.Endothelial dysfunction resulting from chronic stress (e.g., smoking, diabetes, hypertension) leads to the decreased bioavailability of nitric oxide.High circulating levels of LDL infiltrate the damaged endothelium and lead to the development of oxidized LDL. Monocytes are attracted and result in the accumulation of macrophages at the site of inflammation.The accumulation of fat in macrophages leads to the development of foam cells and the deposition of these foam cells in the endothelium, leading to the development of fatty streaks, which are the first signs of atherosclerosis visible without magnification.The resulting inflammation triggers migration of smooth muscle cell across the vascular media, as well as their accumulation and proliferation.The ensuing process of smooth muscle cell proliferation, the release of growth factors and the elaboration of the extracellular matrix results in the development of fibrous plaque.The eventual outcome of the process of atherosclerosis is the development of fibrous plaque, which, if it becomes unstable with the denudation of the endothelium and plaque rupture, can cause an acute thrombotic event.

## 2. Screening

The REACH (REduction in Atherothrombosis for Continued Health) registry was an international prospective cohort study involving 68,236 patients with one of established atherosclerotic diseases (PAD), coronary artery disease (CAD), cerebrovascular disease (CVD), or the presence of three or more major risk factors for atherosclerotic diseases. In the study, 5.35% of patients with pre-existing PAD were found to develop a major cardiovascular event (cardiovascular (CV) death, myocardial infarction (MI), or stroke) at one year, and 21% of PAD patients experienced the composite end point of CV death, MI, or stroke (as opposed to 15.20% and 14.53% for those with CAD and CVD, respectively) [29]. ABI has been noted to be an independent risk factor for adverse cardiovascular outcomes. In the Cardiovascular Health Study, ABI values ≤ 0.6 (HR (hazard ratio) 1.82), 0.61–0.7 (HR 2.08), 0.71–0.8 (HR 1.8), and 0.81–0.9 (HR 1.73) were associated with higher all-cause mortality compared to ABI values of 0.91–1.0 (HR 1.40) [30]. With these data, the idea of earlier diagnosis of PAD through screening is thought to not only prevent PAD progression and the development of complications but also identify these patients who are otherwise at a higher risk of adverse CVD and CAD, implementing the aggressive treatment of risk factors and potentially preventing these adverse atherothrombotic events.

The ACC/AHA guidelines recommend screening for PAD in high-risk patients such as those aged ≥65 years old or those aged 50–64 years old with traditional risk factors [1]. The United States Preventive Services Task Force (USPSTF) guidelines in 2018 concluded that the current evidence is insufficient to recommend PAD screening with ABI in asymptomatic adults [31].

## 3. Presentation

The classic symptom described in PAD is intermittent claudication (IC), which is described as exertional leg pain that is resolved within 10 min of rest. While this has been classically associated with PAD, studies have reported that adults with PAD have a broad range of symptoms besides IC [6,32,33]. Genuine claudication is pain that is not present at rest and begins after a set reproducible amount of distance walked, before being resolved with rest [34]. The PARTNERS (PAD Awareness, Risk, and Treatment: New Resources for Survival) study noted that classic IC is present only in 3.6% of patients with newly diagnosed PAD and 19.5% of patients with established prior PAD. In fact, 54% and 26% of patients with established PAD have atypical leg symptoms or no symptoms at all, respectively [35]. Another US Study evaluated 460 patients sampled from noninvasive laboratories in Chicago and reported a wide range of symptoms other than classic IC. They reported IC symptoms in only 32% of patients and no symptoms or atypical symptoms in in 20% and 48%, respectively [32]. It is important to differentiate leg pain in PAD from other vascular and neurological causes [34].

However, in cases of conventional IC, the anatomic site of presentation follows the suite of the arterial territory of insult, with the concomitant loss of relevant pulses [36]. Aortoiliac claudication presents with hip claudication and feeble or absent femoral pulses. Erectile dysfunction is noted in men with this site of PAD. Similarly, femoral and popliteal disease cause thigh and calf symptoms, and tibial and peroneal disease cause ischemic pain in the foot region [36].

Another rare but serious symptom of PAD is ischemic rest pain, characteristically occurring in the region of the toes or metatarsal heads [37]. Critical limb ischemia (CLI) is denoted as arterial insufficiency due to PAD that results in poor tissue perfusion at rest and presents with the symptoms of rest pain, ulcers, and/or gangrene [36]. Accompanying evidence of low ABI (<0.9) existing with these symptoms helps to distinguish between pain and ulcers due to PAD from other causes such as diabetes and neuropathy [5]. Another distinction that needs to be made is that between CLI and acute limb ischemia, the latter of which is a more rapid or sudden (generally < 2 weeks) process due to the decrease in limb perfusion and subsequent threat to the viability of the limb from pallor, pulselessness, and paralysis [1,38]. CLI is reported to develop in 5–11% cases of PAD, with the incidence being higher in African Americans, Native Americans, and those with co-morbid diabetes mellitus and chronic kidney disease [2,36,39].

The overall rate of complications depends on the progression and extent of PAD, with CLI patients noting the highest risk. CLI patients have been reported to have a 30% one-year risk of amputation and a 50% five-year risk of all-cause mortality [36]. In comparison, patients with stable PAD are reported to have a 20% risk of five-year all-cause mortality [36]. In a retrospective study of 186,338 patients with PAD, the overall rate of all-cause mortality was almost twice as high in patients who underwent amputation compared to those who did not undergo amputation (6.9% vs. 13.5%, *p* < 0.001) [40].

Subclinical atherosclerosis in a middle-aged population, as observed via noninvasive vascular ultrasound, has been reported in the Progression of Early Subclinical Atherosclerosis (PESA) study [41]. PESA is an ongoing prospective cohort study evaluating 4184 middle-aged asymptomatic participants over a period of 19 (2010–2029) years for the development of atherosclerosis. The study utilizes a comprehensive evaluation strategy for the diagnosis of atherosclerotic plaques, such as 2D and 3D vascular ultrasound, noncontrast computed tomography (CT), ^18^fluorodeoxy-glucose position emission tomography (^18^FDG PET)-magnetic resonance imaging, and cardiac magnetic resonance imaging (MRI), along with the baseline clinical interviews, blood tests, and senescence biomarkers in blood.

PESA data noted that even in patients with a coronary calcium score (CAC) of zero or otherwise low cardiovascular risk, about 60% of participants has subclinical plaques at other cardiovascular sites. In fact, the plaques were most frequent in the ilio-femoral territories, and, indeed, the absence of ilio-femoral plaques was strongly associated with the absence of atherosclerosis at other vascular sites. Thus, noninvasive vascular ultrasound screening for ilio-femoral atherosclerotic plaques may be useful as a widespread screening tool for subclinical atherosclerosis.

## 4. Prevention of Major Adverse Cardiovascular Outcomes in PAD

As described earlier in this review, ABI < 0.9 or PAD portend to poor cardiovascular outcomes for patients with an increased risk of all-cause mortality and a higher risk of cardiovascular diseases, such as CAD, stroke, abdominal aortic aneurysm, and even heart failure [1,42,43]. These can be partially ascribed to underlying atherosclerotic risk factors that are otherwise responsible for the development of CAD. Despite this, PAD has been independently attributed to higher rates of future adverse cardiovascular outcomes and death [44]. Additionally, even when adjusting for underlying atherosclerosis and the risk of major adverse cardiovascular events (MACE) attributed to atherosclerosis, patients with PAD are reported to have higher rates of future MACE. Crique et al. conducted the Multi-Ethnic Study (MESA) involving 6647 patients without evidence of clinical CVD at baseline and evaluated the biomarkers of subclinical cardiovascular disease, including ABI. The authors reported an elevated risk ratio of 1.77 for cardiovascular events in patients with ABI < 1.0 (*p* < 0.001), despite adjusting for other underlying risk factors (viz. smoking, body mass index, diabetes, blood pressure, dyslipidemia, high-sensitivity C-reactive protein, d-dimer, interleukin-6, etc.) [45].

These data speak to an urgent need to identify PAD as a CAD risk-equivalent and not as a subset classification under the aegis of major CAD outcomes. Additionally, this highlights the need for more stringent risk factor modification and treatment.

### 4.1. Lifestyle Interventions

#### 4.1.1. Smoking Cessation

Smoking cessation is the greatest modifiable risk factor in the treatment of atherosclerotic diseases, including PAD. Smoking has been implicated in causing endothelial cell injury [46]. Smoking causes endothelial dysfunction and the subsequently increased permeability of the endothelium, tipping the redox balance of the cells toward increased reactive oxygen species and oxidative damage [47,48,49,50].

The Atherosclerosis Risk in Communities (ARIC) study followed 13,335 participants without incident CAD, PAD, or stroke for 26 years and quantified the participants’ risk of developing these three major atherosclerotic diseases based on four smoking parameters (pack-years, duration, intensity, and cessation). The investigators reported the largest effect size for reductions in incident PAD development in all smoking parameters, with an almost four-fold higher incidence of PAD in participants who smoked for more than 40 pack-years. Additionally, an increased risk of atherosclerotic diseases persisted for smoking cessation up to 30 years, and only after a cessation duration > 30 years did the risk of PAD become comparable to that faced by individuals who had never smoked. However, for a period of smoking cessation of >5 years, PAD showed a lower risk of incidence compared to CAD and stroke [51]. The literature’s review studies have shown a strong correlation between smoking and incidence of PAD, with a two-to-three-fold increase in the prevalence of symptomatic PAD in active and former smokers compared to individuals who had never smoked. A dose–response relationship between amount of smoking and PAD was also noted [50].

Smoking cessation is a Class IA recommendation per ACC/AHA guidelines [52]. Direction and verbal motivation by a physician have been reported as positive factors aiding patients in smoking cessation [53]. Varenicline, an α4β2 nicotinic acetyl choline receptor agonist, has been studied for reducing cravings after smoking cessation. A randomized, double-blind, and placebo-controlled trial assessing the safety and efficacy of varenicline in sustaining smoking cessation in 714 patients with stable cardiovascular disease showed increased abstinence from tobacco in the varenicline group. Furthermore, the varenicline group did not show increased cardiovascular mortality, all-cause mortality, or significant adverse effects compared to the placebo groups [54]. Bupropion is another smoking cessation therapeutic option. In total, 629 smokers with CVD were randomized to receive bupropion or placebo, and the study drug cohort reported almost double the rate of abstinence at the 7-week follow-up point [55]. Another Danish nation-wide cohort study compared cardiovascular outcomes in patients treated with varenicline to those treated with bupropion and reported no increased risk of such events in the varenicline cohort [56]. Nicotine replacement therapy (NRT) is also recommended for aiding smoking cessation. A recent network meta-analysis reported no significantly increased risk of cardiovascular toxicity with NRT [57]. Similar studies conducted in the past have reported a small risk of cardiovascular health issues with NRT [58,59].

#### 4.1.2. Diet

Diet has been long reported as playing a major role in supporting lifestyle interventions for preventing the development and progression of coronary artery disease; however, such data are scarce for peripheral artery disease [60,61,62]. A recent Canadian systematic review analyzed forty observational studies and one randomized controlled trial to evaluate the role of diet in PAD. The study, while remarking on the paucity of data for PAD, noted that the currently available data indicated the slightly protective role of the Mediterranean diet [63]. PREDIMED was a multi-center randomized controlled trial conducted in Spain that evaluated the role of the Mediterranean diet in PAD development over a median follow-up period of 4.8 years. The investigators reported a lower incidence of PAD in the Mediterranean diet cohorts [64].

## 5. Current Guidelines on Diagnosis and Treatment of Peripheral Arterial Disease

Both the AHA/ACC [52] and the ESC (European Society of Cardiology)/ESVS (European Society for Vascular Surgery) [65] guidelines recommend resting ABI as the first-line therapy in the diagnosis of PAD as a Class I recommendation, with follow-up imaging playing a role in lesion localization and assessing the degree of stenosis. While the European guidelines do not comment on screening, the AHA/ ACC recommends screening in individuals ≥ 65 years of age, 50 to 64 years old with risk factors or family history of PAD, and ≤50 years old with diabetes and at least one other risk factor, as well as any individual with pre-existing vascular disease. The utility of exercise treadmill ABI is another difference between the two guidelines in terms of diagnosing PAD, where the AHA/ ACC, as a Class I recommendation, have to conduct exercise treadmill ABI on patients with exertional nonjoint-related leg symptoms with normal or borderline resting ABI. The European guidelines have no recommendation regarding the role of exercise treadmill ABI in the diagnosis of PAD.

Although both guidelines agree that multidisciplinary vascular teams are needed, the AHA/ ACC guidelines appear to place greater emphasis on this aspect of care by more clearly defining the roles and members of the team.

The ACC/AHA and ESC/ESVS guidelines have many points of agreement regarding the medical management of PAD, such as the importance of blood pressure control, specifically with angiotensin-converting enzyme inhibitors (ACEi)/angiotensin receptor blockers (ARB) as Class IIA and IIB therapy recommendations, respectively. They also both have a Class IA recommendation for the use of statin therapy for all patients with PAD, and they agree that in diabetics, tight glycemic control is necessary.

Both guidelines speak to the role of antiplatelet monotherapy with either aspirin or clopidogrel in symptomatic patients. However, while the European guidelines suggest that clopidogrel is superior to aspirin based on the results of the CAPRIE (Clopidogrel Versus Aspirin in Patients at Risk of Ischemic Events) trial, the U.S. guidelines make no preference between the two agents. Another area of disagreement between the two guidelines is the role of antiplatelet therapy in treating asymptomatic patients [66]. The U.S. guidelines state that there is a role for antiplatelet therapy in reducing cardiovascular events based upon ABIs that are abnormal (Class IIA) and borderline (Class IIB), while the European guidelines recommend against using antiplatelet therapy in asymptomatic individuals.

Another point of contention is the role of dual antiplatelet therapy (DAPT). The European guidelines state that DAPT can be considered for the first month only after percutaneous and surgical revascularization (Class I), whereas the U.S. guidelines include long-term DAPT in symptomatic patients as a IIB recommendation.

As touched on previously, vorapaxar in combination with antiplatelet therapy is a IIb recommendation in the U.S. guidelines; however, there is no such recommendation regarding vorapaxar in the European guidelines. 

In terms of symptom management, the U.S. guidelines have the use of cilostazol as a Class I recommendation for symptomatic patients, whereas the European guidelines do not recommend cilostazol, as they report that there is limited efficacy, as well as that it carries a risk of adverse effects.

Neither guideline comments on the utility of direct oral anticoagulants (DOACs) in studies such as the COMPASS (Cardiovascular Outcomes for People Using Anticoagulant Strategies) and Voyager were released after the guidelines were published. 

Both guidelines agree that emergent clinical evaluation and treatment are needed in suspected cases of acute limb ischemia (ALI). While the U.S. guidelines stratify patients based on the presence or absence of pulse, the European guidelines suggest that early assessment of symptom severity should guide the clinical approach. Another key difference between these guidelines can be seen in the strategies used for revascularization. The U.S. guidelines favor catheter-based thrombolysis (Class I) over percutaneous mechanical thrombectomy (Class IIA), surgical thrombectomy (Class IIA), and ultrasound-accelerated catheter-based thrombolysis (IIB). The European guidelines recognize that although thrombolysis is likely more appropriate in less severe cases of ALI, there is no clear superiority compared to open intervention. They also make a point to recognize the importance of treating the culprit lesion with either endovascular therapy or open surgery after thrombus removal.

One area in which the European and U.S. guidelines are largely similar is the management of critical limb ischemia, where revascularization is the primary treatment (Figure 2). The main difference between the two is that the European guideline utilizes the WIFI classification, which scores patients based on wound, ischemia, and foot infection. The American guidelines have no such classification system for PAD.

## 6. Lipid-Lowering Therapies

### 6.1. Role of Lipids in Peripheral Arterial Disease

PAD results from the atherosclerotic obstruction of vessels of the lower extremity, typically aortoiliac to pedal arteries [67]. In the past, atherosclerosis was envisaged as a more conventional proliferative process, initially instigated by endothelial injury from oxidative stress with an ensuing platelet aggregative, which, in turn, signaled a proliferative response in endothelial smooth muscle cells via platelet-derived growth factor. This process would eventually form an atheroma that resulted in partial-to-complete vessel obstruction. With the more recent recognition of the role of immune cells and regulators in atherosclerosis and the advent of gene-targeting technology, inflammatory pathways have been heavily implicated in the pathogenesis of atherosclerosis [68]. LDL particles from the bloodstream associate with the proteoglycans in the extracellular matrix and are oxidized (oxLDL) by the reactive oxygen species and enzymes. These activated lipid molecules induce endothelial cell activation and the expression of many vascular cell adhesion molecules, which subsequently bind T-cells and monocytes [69]. Monocytes then differentiate into macrophages, which engulf the LDL particles and form foamy macrophages. Further pathogenesis involves T-helper cells interacting with these macrophages and the release of inflammatory mediators. These inflammatory mediators instigate macrophage activation, the production of proteases, and growth factors, culminating in further fibrosis and plaque formation [70].

The Women’s Health study noted a positive correlation between the elevated small particle LDL (LDL-P) and incident PAD. Also, elevated triglycerides and lower levels of high-density lipoproteins (HDL) have also been observed as significant risk determinants [71,72]. With these data in hand, lipid-lowering therapies are the first line in the management of adverse limb and cardiovascular outcomes in PAD. The literature has repeatedly reported that lipid-lowering therapy, notably statin therapy, has historically been lower in PAD, even after revascularization, compared to patients with CAD [73].

### 6.2. Statins

Statins are a longstanding lipid-lowering therapy with years of data underscoring their safety and efficacy. The cholesterol-lowering effects of statins are due to the competitive inhibitor of the 3-hydroxy 3-methylglutaryl-coenzyme A (HMG Co-A) reductase, thus interrupting the rate-limiting step in cholesterol synthesis [74]. In addition, the other direct effects of HMG Co-A inhibition are inhibition in the hepatic synthesis of apolipoprotein B-100 and increasing apo-B/E receptors [74]. Additionally, statins have been observed to exert antioxidant effects through a reduction in hypercholesterolemia-mediated oxidative stress on the endothelium, the inhibition of endothelial generation of superoxide, the preserving activity of antioxidant enzymes such as superoxide dismutase, and, lastly, the binding of phospholipids on the surfaces of lipoproteins and prevention of lipoprotein oxidation. Statins have also been noted to exert these pleiotropic effects, which improve the cellular function outside of its lipid-lowering effect [75].

Cholesterol treatment trialists conducted a meta-analysis of twenty-six clinical trials involving at least 1000 participants, with main effect of the intervention being LDL reduction. The authors aimed to study the effect of LDL reduction on MACE. The authors show a positive correlation between LDL lowering and a reduction in MACE. However, PAD-related outcomes for major adverse limb events include worsening claudication, critical limb ischemia, peripheral vascular revascularization, or amputation [76].

The ACC/AHA 2016 guidelines confer Class IA recommendation for statin therapy in those with diagnosed PAD. However, the guidelines do not stipulate a LDL cut off, as if available in patients with adverse coronary events (<70 mg/dL) [52]. Multiple studies have been conducted that report the lipid-lowering effects of statins and concomitant improvement in the MACE outcome profile. Some of these studies have reported MALE as a subset analysis. For example, the Scandinavian Simvastatin Survival Study (4S) is one of the initial trials elucidating the role of statins in improving the mortality and major coronary events in patients with established coronary artery disease. While this study reported outcomes related to an improvement in MACE and provided first of its kind data using the subgroup analysis of women as a separate investigational cohort, the trial did not include MALE as an outcome measure [77]. The Heart Protection Study (HPS) enrolled 20,536 patients with coronary artery disease, other occlusive atherosclerotic diseases, or diabetes and randomized them to receive simvastatin versus placebo over a five-year treatment period. The study reported a significant reduction in the major vascular event (coronary death, nonfatal myocardial infarction (MI), fatal or nonfatal stroke, or any revascularization) rate in the study group (19.2% vs. 25.2%, *p* < 0.001). For additional subgroup analyses of the patients with pre-existing PAD, with or without prior CAD, simvastatin significantly reduced the major vascular event rate (26.4% vs. 32.7%, *p* < 0.0001) [78]. The subgroup analysis from the EUCLID Trial also noted that prior statin use was associated with a lower risk of major amputation (HR 0.52, *p* < 0.001) [79].

Westin and colleagues devised a retrospective chart review of 380 patients with CLI who underwent either a diagnostic angiography or therapeutic endovascular intervention and evaluated associations between statin use and MACE and MALE. The authors reported lower one-year MACE (stroke, MI or death) rates in statin users (HR 0.53, CI (Confidence Interval) 0.28–0.99). Also, major amputation or death rates were significantly lower in the statin cohort (HR 0.53, CI 0.35–0.98), with improved rates of lesion patency observed during infrapopliteal angiography in the statin users [80]. Another study of CLI patients was devised by Stavroulakis et al. using the patients enrolled in the First-Line Treatments in Patients With Critical Limb Ischemia (CRITISCH) registry. Patients on statins were observed to have a lower risk of amputation free survival (AFS) (HR 0.45, *p* < 0.001); however, statin therapy was not associated with reduced amputation rates (HR 1.02, *p* = 0.922) [81]. In a 2020 meta-analysis of 19 studies reporting outcomes on statin treatment in patients with CLI, the authors reported 25% and 38% relative risk reductions in the amputation rates (HR 0.75, CI 0.59–0.95) and overall mortality (HR 0.62, CI 0.52–0.75) [82].

Recent studies have reported that elevated lipoprotein (a) levels are associated with a higher incidence of MALE. In a recent single-center retrospective study, 16,513 hospitalized patients from 2000 to 2020 were evaluated. The one-year incidence of MALE outcomes was higher in those with higher Lp(a) (lipoprotein a) levels compared to the overall population (HR 4.54% vs. 2.44%, *p* = 0.01). The authors also reported that the high and very high Lp(a) levels were independent risk factors for MALE [83]. 

### 6.3. Proprotein Convertase Subtilisin/Kexin Type 9 (PCSK9) Inhibitors

PCSK9 is a proprotein convertase that binds to and degrades the cell surface LDL receptors. PCSK9 inhibitors are monoclonal antibodies that bind to PCSK9 and inhibit LDL receptor degradation and increase the clearance of LDL cholesterol from the blood stream [84]. The Food and Drug Administration (FDA) of the United States has approved two PCSK9 inhibitors: alirocumab (Praluent) and evolucumab (Repatha).

FOURIER (Further Cardiovascular Outcomes Research with PCSK9 Inhibition in Subjects with Elevated Risk) was a randomized controlled trial involving 27,564 patients with established atherosclerotic disease already on statin therapy, who were randomized to receive evolucumab vs. placebo over a median follow-up of 2.2 years. Of the overall study population, 13.2% (642) patients had prior PAD diagnosis. The study primary end point was a composite of cardiovascular death, myocardial infarction, stroke, hospital admission for unstable angina, and coronary revascularization. The key secondary end point was a composite of cardiovascular death, MI, and stroke. While evolucumab consistently improved the primary and secondary end points in the overall study population compared to the placebo, the results were even more striking for the PAD cohort. The key secondary end point for those with PAD noted more significant improvement with evolucumab (HR 0.73, *p* = 0.0040 vs. HR 0.81, *p* < 0.0001). The authors also reported that due to the inherent higher risk profile of the patients with PAD, this cohort was found to have higher absolute risk reductions in the primary composite end point (3.5% in PAD vs. 1.4% in those without PAD) [85,86].

ODESSEY OUTCOMES was another randomized controlled trial evaluating cardiovascular outcomes in 18,924 patients with recent acute coronary syndromes (within 1–12 months of trial enrollment). The primary end point was a composite of death from coronary heart disease, nonfatal MI, fatal, or nonfatal stroke, or unstable angina requiring hospitalization. The patients were followed for a median period of 2.8 years. The trial results noted overall improvement in the primary end point in the alirocumab group compared to the placebo group (HR 0.85, *p* < 0.001). However, investigators in the original trial did not report segregated data on PAD outcomes or complications [87]. In a prespecified analysis of the trial used to ascertain the impact of alirocumab on PAD events and the association of these events with Lp(a), investigators reported that alirocumab reduced the risk of PAD events (CLI, limb revascularization or amputation for ischemia) (HR 0.69, *p* = 0.004). It was also observed that the risk of PAD events positively correlated with higher Lp(a) levels and was improved by alirocumab [88].

### 6.4. Ezetimibe

Ezetimibe acts by inhibiting cholesterol absorption in the intestine. The drug, as well as its secondary metabolites, inhibit the cholesterol transporters on the surface of intestinal brush border cells [89]. Currently, ezetimibe is recommended as an add-on therapy to statins in patients who are unable to reach goal LDL levels with high-intensity statin doses alone [90]. The IMPROVE-IT (IMProved Reduction in Outcomes: Vytorin Efficacy International Trial) trial evaluated the primary composite of cardiovascular endpoints (CV death, MI, stroke, unstable angina leading to hospitalization, coronary revascularization ≥ 30 days post-randomization) in post-acute coronary syndrome patients who received a simvastatin–ezetimibe combination compared to placebo. While the study was notably optimistic, with a 9% reduction (HR 0.91, *p* = 0.007) in the primary end point recorded in the study group, the trial did not include any PAD events in the primary composite end point [91]. Ezetimibe has been paradoxically reported in the literature to cause the progression of peripheral atherosclerosis. West et al. studied the atherosclerotic plaque volume in the superficial femoral artery and re-analyzed the plaque volume in the patients who started taking simvastatin or simvastatin with ezetimibe. The authors reported that atherosclerotic plaque progression was observed in patients who were already on simvastatin, and ezetimibe was started due to LDL-C being ≥80 mg/dL. However, this study is not generalizable given some inherent flaws due to small sample size and low power, as well as high attrition rates, as 23% of patients who were enrolled at initiation dropped out before the study’s conclusion, and lack of a comparison arm [92].

### 6.5. Icosapent Ethyl

Icosapent ethyl (IPE) is a purified ester of eicosapentanoic acid, which was recently shown to improve ischemic outcomes. The likely mechanism of these effects has been hypothesized to be secondary to several pleiotropic effects exerted by IPE [93].

The REDUCE-IT (Reduction in Cardiovascular Events with Icosapent Ethyl-Intervention Trial) trial randomized 8179 patients with triglyceride levels ≥ 135–500 mg/dL to receive 4 g/day IPE vs. placebo. These patients were already on a stable dose of statin. Of note, 688 patients in the study had previously established PAD diagnosis. The primary end point was composite of cardiovascular death, nonfatal MI, nonfatal stroke, coronary revascularization, or hospitalization for unstable angina. The study group reported a 30% reduction in the primary endpoint (HR 0.70, *p* < 0.0001) [94]. In a subset analysis of the trial evaluating the patients with PAD, IPE significantly reduced the total primary composite end points, with results of 112.8 per 1000 patient years with IPE vs. 162.3 per 1000 patient years with placebo (HR 0.68, *p* = 0.03) being recorded [95].

### 6.6. Bempedoic Acid

Bempedoic acid is another newer lipid-lowering agent that has been evaluated for its role in supplementing lipid-lowering effects in concert with statin prescription. Bempedoic acid inhibits the ATP (adenosine triphosphate) citrate lyase enzyme and, thus, interrupts cholesterol synthesis. This, in turn, upregulates the LDL-receptors on the cell surface and improves LDL clearance from the blood stream [96].

The CLEAR (Cholesterol Lowering via Bempedoic Acid [ECT1002], an ACL-Inhibiting Regimen) OUTCOMES trial evaluated the lipid-lowering effect and cardiovascular benefit of bempedoic acid in patients who cannot take the guideline-recommended doses of statins. The primary end point was a composite of cardiovascular death, nonfatal MI, nonfatal stroke, or coronary revascularization. The investigators reported that the bempedoic acid group was found to have a significantly lower rate of primary end point (11.7% vs. 13.3%, *p* = 0.004) [97]. In this trial, 1624 of the 13,970 patients enrolled had previously been diagnosed with PAD. However, a subset analysis of the PAD cohort is not available. Also, the PAD events were not included in the primary end points of the trial.

### 6.7. Inclisiran

Inclisiran is a double-stranded small interfering RNA (siRNA) that inhibits hepatic PCKS9 translation. The European Union approved the use of inclisiran in the treatment of familial hyperlipidemia after the success of the ORION trials [98]. Another randomized controlled trial evaluating the effect of inclisiran on lowering cholesterol was devised by Ray and colleagues. The investigators enrolled 501 patients with elevated cholesterol levels or high atherosclerotic cardiovascular disease risk (ASCVD) (LDL-C ≥ 70 mg /dL for those with high ASCVD risk or LDL-C ≥ 100 mg/dL for those without high ASCVD risk) and already on a stable dose of statin therapy (≥30 days), and they were randomized to receive placebo versus inclisiran at increasing doses. The primary endpoint of the study was LDL-C level at 180 days. At 180 days, the highest LDL-C reduction was observed in the two-dose 300 mg inclisiran regimen with LDL-C level < 50 mg/dL at day 180. Additionally, at day 240, the PCSK9 and LDL-C level remained low in all inclisiran dose study groups. While there are optimistic data supporting the action of inclisiran for LDL-C reduction, further data on its effects in MACE and MALE modification in PAD are not yet available [99].

The HPS-4/TIMI65 ORION 4 trial is currently underway, and it aims to explore the effects of inclisiran on major adverse cardiovascular outcomes (CV death, nonfatal MI, nonfatal stroke, or urgent coronary revascularization) (NCT03705234).

Figure 3 depicts the mechanisms of action of various hyperlipidemia therapies.

## 7. Antithrombotic Agents

The benefit of long-term antithrombotic therapy for the prevention of atherosclerotic cardiovascular disease has been studied in both asymptomatic and symptomatic patients with PAD. In patients with asymptomatic PAD, antiplatelet therapy is a Class IIA recommendation and should be individualized, considering the overall cardiovascular risk and the risk of bleeding [52]. Two studies have investigated the treatment of patients with asymptomatic PAD with aspirin therapy; however, neither study was sufficiently powered to derive a meaningful conclusion [100,101].

In patients with symptomatic PAD, AHA/ACC guidelines currently have a class IA recommendation for antiplatelet monotherapy with aspirin (75–325 mg/day) or clopidogrel (75 mg/day) to reduce MI, stroke, and vascular death [52]. The Antithrombotic Trialists Collaboration (ATC) performed a meta-analysis in which aspirin was the most widely studied antiplatelet and found a 22% reduction in cardiovascular events in patients who suffered from symptomatic PAD compared to placebo [102]. In the Clopidogrel Versus Aspirin in Patients at Risk of Ischemic Events (CAPRIE) trial, clopidogrel (75 mg/day) had a significant advantage over aspirin (325 mg/day) in terms of reducing the risk of a combined outcome of ischemic stroke, MI, or vascular death in symptomatic patients (RR (Relative Risk) 0.76, 95% CI 0.64–0.91) [103]. The Examining Use of Ticagrelor in Peripheral Artery Disease (EUCLID) study also studied symptomatic patients and found no significant difference between ticagrelor and clopidogrel in terms of preventing adverse cardiac events (HR 1.02, 95% CI 0.92–1.13) or acute limb ischemia (HR 1.03, 95% CI, 0.79–1.33) [104].

Dual antiplatelet therapy (DAPT) with aspirin and clopidogrel in patients with symptomatic PAD is not well established and is considered to be a IIb recommendation [54]. The Clopidogrel for High Atherothrombotic Risk and Ischemic Stabilization, Management, and Avoidance (CHARISMA) trial compared DAPT to clopidogrel plus aspirin versus aspirin alone in patients with symptomatic PAD and found no significant difference in adverse cardiovascular events (HR 0.85, 95% CI 0.66–1.08 [105]. The Prevention of Cardiovascular Events in Patients With Prior Heart Attack Using Ticagrelor Compared to Placebo on a Background of Aspirin-Thrombolysis in Myocardial Infarction 54 (PEGASUS-TIMI 54) trial compared DAPT to ticagrelor and aspirin versus aspirin alone in patients with symptomatic PAD and prior MI and found significant reductions in adverse cardiovascular (ARR (absolute risk reduction) 4.1%, 95% CI 1.07–9.29%) and limb events (HR 0.65, CI 95% 0.44–0.95) [106].

DAPT in patients with PAD after lower extremity revascularization is also a IIb recommendation, although few small RCTs have shown benefit after both endovascular revascularization and prosthetic bypass grafts [52,107,108]. More recent data from the Vascular Outcome Study of Aspirin Along with Rivaroxaban in Endovascular or Surgical Limb Revascularization for Peripheral Artery Disease (VOYAGER PAD) study demonstrate a significant benefit of low-dose rivaroxaban (2.5 mg/ day) plus aspirin versus aspirin alone. The benefit included improvements in composite death from cardiovascular causes, ischemic stroke, and MI, as well as a reduction in acute limb ischemia and major amputation after revascularization (HR 0.85, 95% CI 0.76–0.96) [109].

Vorapaxar is an antagonist protease-activated receptor-1, the primary receptor for thrombin on platelets, vascular endothelium, and smooth muscle. The Trial to Assess the Effects of SCH 530348 in Preventing Heart Attack and Stroke in Patients with Atherosclerosis (TRA2P-TIMI 50) revealed that although vorapaxar did not reduce the risk of cardiovascular death, MI, or stroke in patients with PAD, it significantly reduced the rates of acute limb ischemia (HR 0.58, CI 95% 0.39–0.86) and peripheral revascularization (HR 0.84, 95% CI 0.73–0.97) [110].

In the Warfarin Antiplatelet Vascular Evaluation (WAVE) trial, patients with symptomatic PAD were randomized to therapeutic warfarin or acenocoumarol (target internationalized normal ratio [INR] 2.0–3.0) plus antiplatelet therapy with either aspirin, clopidogrel, or ticlopidine versus antiplatelet therapy alone. The addition of anticoagulation therapy did not reduce MACE, but it increased the risk of life-threatening bleeding (RR 3.41, 95% CI 1.84–6.35) [111]. The Cardiovascular Outcomes for People Using Anticoagulation Strategies (COMPASS) trial randomized 27,395 patients with various forms of chronic atherosclerotic cardiovascular disease to rivaroxaban 5 mg twice daily versus rivaroxaban 2.5 mg twice daily plus aspirin versus aspirin alone. The trial was stopped early due to the superiority of low-dose rivaroxaban plus aspirin over both of the other therapies in terms of reducing cardiovascular death, stroke, or MI [112]. A sub-analysis of COMPASS found that low-dose rivaroxaban plus aspirin, when compared to aspirin alone, reduced MACE (HR 0.72, 95% CI 0.57–0.90) and MALE (HR 0.54, 95% CI 0.35–0.82); however, it had an increased risk of major bleeding (HR 1.61, 95% CI 1.12–3.31) [113]. Figure 4 describes the various antithrombotic agents and their mechanisms of action.

## 8. Diabetes Management

Diabetes mellitus is known to be a contributor to the atherosclerotic pathophenotype, and it has been theorized that diabetes would further exacerbate PAD. While this might be so, the evidence regarding the effect of diabetes on MACE and MALE outcomes in PAD is rather incomplete.

In an observational study used to account for cardiovascular events (MI, stroke, and CLI) in diabetics, Camafort et al. followed 974 patients with type 2 diabetes and stable cardiovascular disease (CAD, PAD, or cerebrovascular disease) for a period of 14 months. The authors noted that intensive glycemic control (Hemoglobin A1c (HbA1c) < 7.0%) resulted in fewer adverse cardiovascular events as compared to those with HbA1c > 7.0%. However, upon subgroup analysis, this effect was only significant for the patients with coronary artery disease (HR 0.4, CI 0.2–0.8) and did not hold true for those with PAD (HR 0.8, CI 0.5–1.3) [114]. In the Action to Control Cardiovascular Risk in Diabetes (ACCORD) trial, Goldman et al. randomized 9985 patients with type 2 diabetes mellitus into intensive (HbA1c < 6.0%) and standard (HbA1c 7–7.9%) glycemic control groups and evaluated the lower extremity amputations (LEA) over the course of the trial and at the post-trial follow-up. The investigators reported lower risk of LEA in the intensive glycemic control group compared to the standard group; after 3.7 years of intensive glycemic control, this treatment strategy was noted to have protection against LEA. This trial has 376 patients with prior established peripheral arterial disease; however, subgroup analysis of the primary study outcome for the PAD patient cohort was not performed, providing no certain data for this disease population [115].

Sodium Glucose Co-Transporter-2 (SGLT-2) Inhibitors are a Class IA recommendation in the guideline-directed medical therapy HFrEF (heart failure with reduced ejection fraction) and Class IIA recommendation in HFpEF (heart failure with preserved ejection fraction) for reductions in heart failure hospitalization and CV mortality [116]. However, a similar cardioprotective role of SGLT2 inhibitors has not been reported in PAD. EMPA-REG OUTCOME examined the role of empagliflozin in reducing cardiovascular outcomes (CV death, nonfatal MI, or nonfatal stroke) in type 2 diabetics with high cardiovascular risk. The study group noted a significant reduction in the primary composite outcome, but the subgroup analysis for PAD cohort did not reach significance (HR 0.94, CI 0.47–1.88, *p* = 0.53) [117]. The DECLARE-TIMI 58 (Dapagliflozin Effect on Cardiovascular Events–Thrombolysis in Myocardial Infarction 58) trial studied the effect of dapagliflozin on MACE (CV death, nonfatal MI, or nonfatal stroke) in type 2 diabetics. While dapagliflozin resulted in reductions in CV death and hospitalization for heart failure (HR 0.83, *p* = 0.005), subgroup analysis for PAD patient cohort was not reported in the trial [118]. The CANVAS (Canagliflozin Cardiovascular Assessment Study)-R (Renal) and CANVAS Program were devised to evaluate the primary outcomes of CV death, nonfatal MI, and nonfatal stroke in type 2 diabetics taking 100 mg or 300 mg of canagliflozin daily versus placebo. Canagliflozin reduced the primary outcome compared to placebo (HR 0.86, *p* for superiority 0.02, *p* for noninferiority < 0.0001). Interestingly, a higher rate of lower extremity amputation was observed compared to placebo (HR 2.85, *p* < 0.001). However, upon subgroup analysis in the PAD patient cohort in the trial, the lower extremity amputation rate did not reach significance (HR 1.38, CI 0.8–2.4) [119].

The Liraglutide Effect and Action in Diabetes: Evaluation of Cardiovascular Outcome Results (LEADER) trial investigated the effect of liraglutide on MACE (CV death, nonfatal MI or nonfatal stroke) in patients with type 2 diabetes. The study reported a significantly lower rate of MACE in the liraglutide study group (HR 0.87, *p* for superiority 0.01, *p* for inferiority < 0.001). However, the trial did not report PAD subgroup data [120]. A similar trial evaluating the role of semaglutide in type 2 diabetics was the SUSTAIN (Trial to Evaluate Cardiovascular and Other Long-Term Outcomes with Semaglutide in Subjects with Type 2 Diabetes) trial. This trial also reported a lower rate of adverse cardiovascular outcomes (CV death, nonfatal MI, or nonfatal stroke) in the semaglutide group compared to placebo (HR 0.74, CI 0.58–0.95, *p* for noninferiority to placebo < 0.001) [121]. However, the SUSTAIN trial investigators also did not report PAD-subgroup-specific data. The Semaglutide Against Placebo in People With Peripheral Arterial Disease And Type 2 Diabetes Mellitus (STRIDE) clinical trial is currently underway, with the first results from the study currently expected in July 2024. The STRIDE clinical trial is investigating the role of semaglutide in patients with type 2 diabetes and symptomatic PAD. The PAD-specific outcomes studied in this trial are the change in maximum walking distance (primary outcome), change in ‘pain-free’ walking distance, and follow-up ABI and Vascular Quality of Life Questionnaire-6 (VascuQoL-6) scores (NCT04560998).

Arya et al. devised a retrospective observational study of 26,799 veterans who underwent PAD revascularization procedures between 2003 and 2014. The authors compared amputation and MALE risk for HbA1c levels using Kaplan–Meir analysis. Cox proportional hazard models were created to evaluate the effects of elevated HbA1c levels on the rates of amputations and MALEs. The authors reported that with incremental levels of HbA1c values, there was a statistically significant higher risk of both amputations and MALEs [122].

## 9. Antihypertensives

Current AHA/ACC guidelines have a Class IA recommendation that antihypertensive therapy should be made available for patients with both hypertension and PAD to reduce the risk of stroke, myocardial infarction (MI), heart failure, and cardiovascular death [52]. Several studies have described a J-shaped pattern for cardiovascular events as systolic blood pressure (SBP) is lowered beyond an apparent ideal range. In a post hoc analysis of the International Verapamil-SR/Trandolapril Study (INVEST), a J-shaped relationship was established in both systolic and diastolic blood pressure (DBP), wherein both lower and higher pressures resulted in adverse cardiac outcomes. They found that the ideal SBP was 135–145 mmHg with a DBP of 60–90 mmHg [100]. The Appropriate Blood-Pressure Control in Diabetes (ABCD) study found that the intense lowering of blood pressure to an average of 128/75 mmHg was associated with a reduction in cardiovascular events in patients suffering from both diabetes mellitus and PAD [123]. In a follow-up study of the Heart Outcomes Prevention Evaluation (HOPE), it was found that ramipril significantly reduced the rates of cardiovascular mortality, stroke, and MI in patients with clinical and subclinical PAD, regardless of ABI, when compared to placebo. The study helped to establish the use of an angiotensin-converting enzyme inhibitor or angiotensin receptor blocker as a Class IIA recommendation for PAD, but it did not report a target blood pressure [52,124].

The above studies present evidence in support of the existence of an ideal SBP inflection point regarding adverse cardiovascular events. Two additional studies present data regarding aggressive blood pressure management in patients with PAD and the theoretical risk of decreased limb perfusion. The first study is a sub-analysis of the EUCLID trial, which confirmed that every 10 mmHg increase in systolic blood pressure (SBP) > 125 mmHg was associated with an increased risk of major adverse cardiac events (HR 1.10, 95% CI 1.06–1.14) and recognized increased risk of major adverse limb events and lower extremity revascularization (HR 1.08, 95% CI 1.04–1.11). However, every 10 mmHg decrease in SBP ≤ 125 was associated with an increase in major adverse cardiac events (HR 1.19, 95% CI, 1.09–1.31) but not in major limb events or lower extremity revascularization [125]. The second study is the Antihypertensive and Lipid-Lowering Treatment to Prevent Heart Attack Trial (ALLHAT), which revealed the existence of an SBP J-curve similar to that previously described for adverse cardiovascular events in lower extremity vascular limb events. In the study, SBP < 120 (HR 1.26, 95% CI 1.05–1.52) and SBP > 160 (HR 1.21, 95% CI 1.0–1.48) were both associated with a higher risk of PAD events compared to SBP of 120–129. Lower but not higher DBP was associated with a higher risk of PAD events for DBP < 60 (HR 1.72; 95% CI, 1.38–2.16). It is important to note that these studies are all retrospective, sub-analyses of trials, the primary focus of which was not, in any case, hypertension management in PAD [126]. Furthermore, the conflicting evidence regarding the existence of a PAD adverse outcome J-curve between these final two studies also warrants further investigation.

## 10. Autologous Stem Cell Therapy

Autologous stem cell therapy, as it pertains to PAD, starts with the isolation of endothelial progenitor cells (EPC’s) from one’s own blood via magnetic bead selection, centrifugation, or one of various adhesion methods. These EPCs can be administered either intramuscularly or intra-arterially and promote angiogenesis, thereby restoring perfusion to distal tissue. Multiple meta-analyses have been conducted regarding this novel approach, and they have shown mixed results. Rigatto et al. published a systematic review in which cell therapy was associated with a 37% (RR 0.63, 95% CI 0.49–0.82) reduction in amputation and a 59% (CI 19–113%) increase in the probability of complete wound healing. 

When only placebo-controlled trials were analyzed, however, the effects were no longer statistically significant [127]. A smaller meta-analysis from Moazzami et al. was unable to draw any conclusions when analyzing data limited by strict inclusion criteria [128]. Xie et al. showed a 41% (RR 0.59, 95% CI 0.46–0.76) reduction in amputation, a 0.13 (Median (MD) 0.13, 95% CI 0.11–0.15) mmHg increase in ABI, and a 73% (RR 1.73, 95% CI 1.45–2.06) increase in ulcer healing [129]. Pu et al. found a 31% (RR 0.69, 95% CI, 0.50–0.94) reduction in amputation, a 0.08 (MD 0.08, 95% CI, 0.02–0.13) mm Hg increase in ABI, and significant improvement in the rest pain score; however, death and ulcer size did not improve [130]. A meta-analysis by Gao et al. showed that stem cell therapy was more effective at increasing the healing rated of ulcers (OR (Odd’s ratio) 4.31, 95% CI 2.94–6.30) and ABI (MD 0.13, 95% CI 0.10–0.17), with a reduction in amputation (OR 0.66, 95% CI 0.42–1.03) and rest pain scores (MD −1.61, 95% CI −2.01–1.20). The side effects of this novel therapy include lower extremity edema, bleeding, cellulitis, proliferative retinopathy, and wound sepsis that results in amputation [131].

## 11. Growth Factor and Gene Therapy

Gene therapy works via the direct delivery of genetic material into cells through viral or nonviral means. Modified viruses, such as adenovirus, adeno-associated virus, and lentivirus, are able to effectively deliver genetic material to target cells and tissue. Nonviral gene therapy refers to plasmids. These are circular double-stranded nucleic acid (DNA) molecules containing a promoter that drives the transcription of a transgene encoding a therapeutic protein of choice. In the context of PAD, gene therapy attempts to increase vascular supply via vasculogenesis, arteriogenesis, and angiogenesis, thereby restoring muscle perfusion and function. Although there are no FDA-approved gene therapies for PAD, plasmid-based therapies for PAD exist, and they are utilized in other parts of the world. Examples of such therapies include Collatagene in Japan and Neovasculgen in Russia and Ukraine. These two agents code for hepatocyte growth factor and vascular endothelial growth factor, respectively [132,133]. While some randomized control clinical trials have found there to be some utility in gene therapy, multiple meta-analyses have been unable to corroborate these results. The first meta-analysis published regarding the utility of gene therapy in PAD was conducted by De Haro et al., reviewing six randomized controlled trials (RCT). They concluded that although there was no difference in mortality, gene therapy showed significant efficacy compared to placebo in critical ischemia (OR 2.20; 95% CI 1.01–4.79) [134]. A meta-analysis by Forster et al. reviewed 17 RCTs and found no clear differences in terms of amputation-free survival, major amputation, and all-cause mortality between those treated with gene therapy and those not treated with gene therapy [135]. Hammer et al. reviewed 12 RCTs and found no differences in terms of mortality, amputations, or ulcer healing [136]. Further studies are clearly warranted before the widespread use of these agents can be recommended. 

## 12. Conclusions

The presence of peripheral artery disease is associated with 3–4 times higher mortality at all Framingham risk levels. Despite the high prevalence, PAD continues to be frequently missed during diagnostics tests. With a cost burden of over 21 billion USD, efforts at increasing the awareness of patients and providers regarding PAD and its associated risks of poor cardiovascular outcomes need to be improved.

In this review, we note that while intermittent claudication is the most classically associated clinical symptom of PAD, the majority of patients with PAD will either be asymptomatic or present atypical symptoms. In light of this knowledge, provider suspicion for PAD is important for the timely diagnosis and treatment of the condition. Atherosclerosis, mainly instigated by atherogenic lipoproteins and associated with plaque formation, progression, and vessel occlusion, is the main pathophysiologic mechanism of PAD. The role of smoking and lifestyle factors in promoting an atherogenic profile and the further progression of the disease has been unequivocally established in the literature.

With this knowledge, smoking cessation continues to be the greatest modifiable risk factor in PAD treatment and the prevention of MACE and MALE. Lipid-lowering agents are the mainstay of controlling hyperlipidemia, which also improves cardiovascular and peripheral limb outcomes. Statins have years of robust data supporting their efficacy and safety in the treatment of hyperlipidemia and improvement in CV outcomes. The efficacy of statins is also attributed in part to their pleiotropic effects, outside of their lipid-lowering ability. Additionally, the relatively newer PCSK9 inhibitors are potent lipid-lowering agents that have shown strong evidence of a potent LDL-C reduction ability. Data on the role of evolucumab in PAD outcomes have been reported in clinical trials.

The roles of antithrombotic agents has been explored and established in clinical trials. The roles of antihyperglycemic agents and the maintenance of a HbA1c level below 8 have been positively correlated with improved PAD outcomes. Similarly, the roles of ACE inhibitors have also been established in randomized controlled trials for improving MACE outcomes.

The use of autologous stem cells and growth factor therapy are novel therapeutic avenues being explored for the treatment of PAD and prevention of adverse cardiovascular and limb outcomes. However, while promising, the data on these therapeutic interventions are mainly observational, and future clinical trials are needed to establish their safety and efficacy.

## Figures and Tables

**Figure 1 biomedicines-11-03157-f001:**
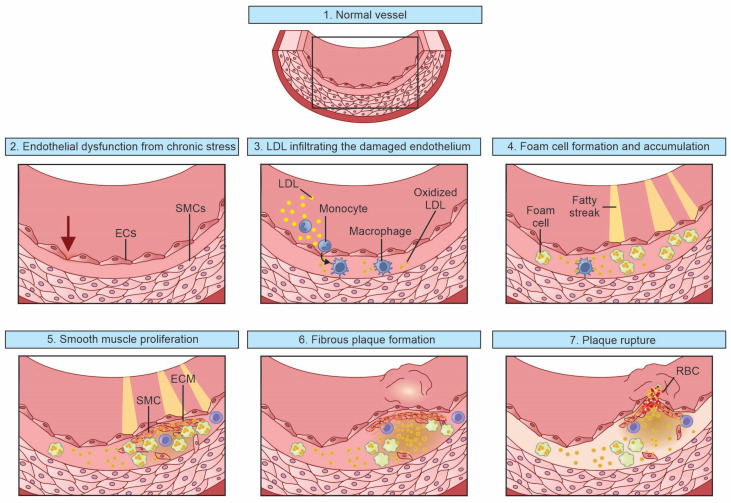
Overview of the pathophysiologic mechanisms involved in the formation and ultimate rupture of an atherosclerotic plaque. EC: endothelial cells; ECM: extracellular matrix; LDL: low-density lipoprotein; RBC: red blood cells; SMC: smooth muscle cell.

**Figure 2 biomedicines-11-03157-f002:**
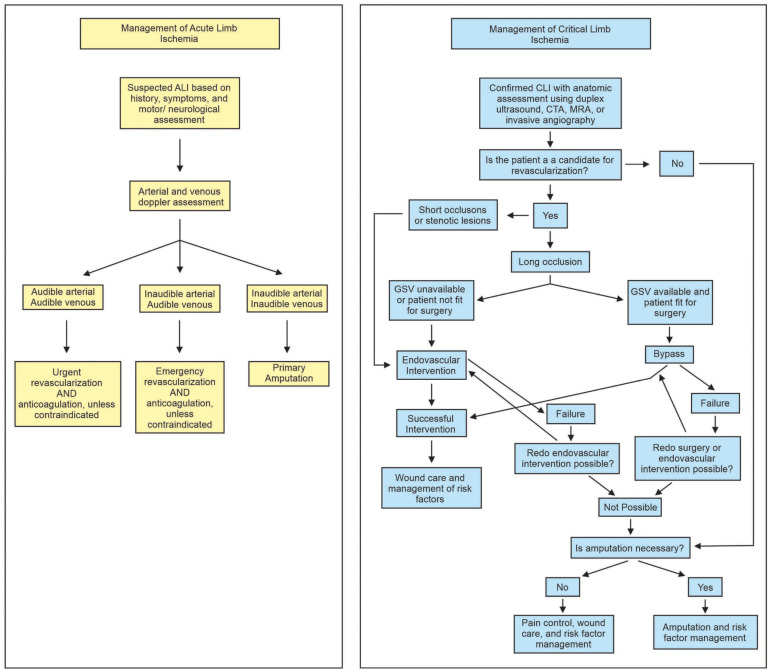
Management of acute limb ischemia (**left**) and chronic limb ischemia (**right**). ALI: acute limb ischemia, CLI: chronic limb ischemia, GSV: great saphenous vein, CTA: computed tomography angiography, MRA: magnetic resonance angiography. Modified and adapted from Gerhard-Herman et. al. [52] and Aboyans et. al. [65].

**Figure 3 biomedicines-11-03157-f003:**
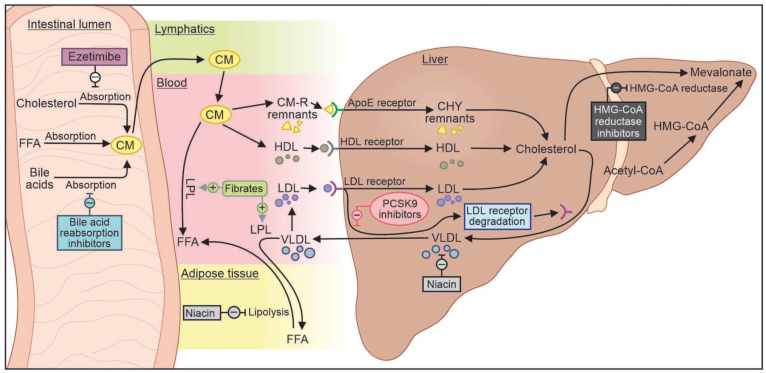
Mechanisms of action of various lipid-lowering therapies used in the management of hyperlipidemia in peripheral artery disease. Acetyl Co-A: acetyl coenzyme A; ApoE: apolipoprotein E; CM: chylomicron; CM-R: chylomicron remnant; FFA: free fatty acids; LPL: lipoprotein lipase; HDL: high-density lipoprotein; HMG Co-A: 3-hydroxy-3-methylglutaryl coenzyme A; LDL: low-density lipoprotein; PCSK9: Proprotein convertase subtilisin/kexin type 9; VLDL: very-low-density lipoprotein.

**Figure 4 biomedicines-11-03157-f004:**
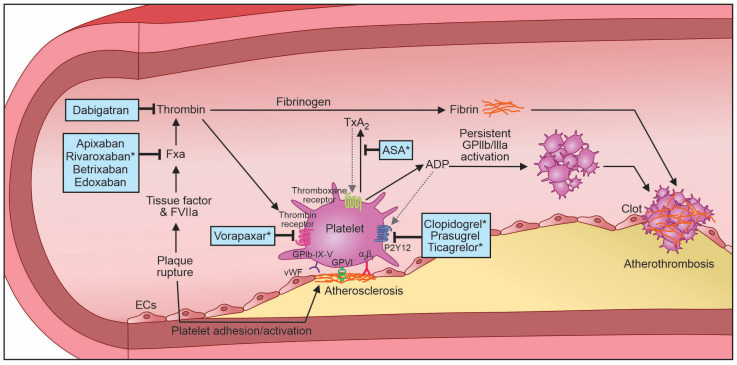
Mechanisms of action of various antithrombotic agents. ADP: adenosine diphosphate; ASA: aspirin; EC(s): endothelial cells; FXa: factor Xa; FVIIa: factor VIIa; GpIb: glycoprotein Ib receptor; GpIIb/IIIa: glycoprotein IIb/IIIa receptor; P2Y12: Purinergic receptor P2Y G-protein coupled 12 protein; TxA_2_: thromboxane A_2_. * FDA-approved antithrombotics for the treatment of peripheral artery disease.

## Data Availability

Not applicable.

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
