# Peer review of "Preventive Therapies in Peripheral Arterial Disease"

_biomedicines, 2023, doi:10.3390/biomedicines11123157_

Round 1

Reviewer 1 Report

Comments and Suggestions for Authors

I have read this review paper on peripheral artery disease with great interest. It is well written, providing interesting overview of biology, epidemiology and partly investigating treatment. 

I have only minor comments:

1) abbreviations should be explained upon mentioning (i.e. ABI).

2) Paper would benefit from the section considering treatment recommendations from at least one relevant international societies (or comparison), not only the review of emerging therapies. This section should focus on medications and interventions.

Reviewer 2 Report

Comments and Suggestions for Authors

Dear Authors

my compliments for this deeply review on atherosclerosis and PAD.

I have only few concerns:

- Figure 2 could be removed

- How did you perfomr this review? which words have you seeked?

- Wich papers or previous review or RCT have you preferred?

- Did you keep in mind the role of ultrasound control of the plaques to assess the level of diseases?

- How did you achieve the success of the terapies?

- Have you entailed the association with cardiovascular diseases?

Please add others fields of future perspectives
